# Circulating Tumor Cells as a Tool of Minimal Residual Disease Can Predict Lung Cancer Recurrence: A longitudinal, Prospective Trial

**DOI:** 10.3390/diagnostics10030144

**Published:** 2020-03-06

**Authors:** Ching-Yang Wu, Chia-Lin Lee, Ching-Feng Wu, Jui-Ying Fu, Cheng-Ta Yang, Chi-Tsung Wen, Yun-Hen Liu, Hui-Ping Liu, Jason Chia-Hsun Hsieh

**Affiliations:** 1Division of Thoracic and Cardiovascular Surgery, Department of Surgery, Chang Gung Memorial Hospital, Linkou 333423, Taiwan; wu.chingyang@gmail.com (C.-Y.W.); maple.bt88@gmail.com (C.-F.W.); ctwen2001@gmail.com (C.-T.W.); foreverairmail@gmail.com (Y.-H.L.); hpliu125@ms21.hinet.net (H.-P.L.); 2College of Medicine, Chang Gung University, Taoyuan 333323, Taiwan; juiing0917@hotmail.com (J.-Y.F.); Yang1946@cgmh.org.tw (C.-T.Y.); 3Department of Medicine, School of Medicine, National Yang-Ming University, Taipei 112304, Taiwan; u502107@yahoo.com.tw; 4Department of Medical Research, Taichung Veterans General Hospital, Taichung 407752, Taiwan; 5Division of Endocrinology and Metabolism, Department of Internal Medicine, Taichung Veterans General Hospital, Taichung 407752, Taiwan; 6Division of Pulmonary and Critical Care Medicine, Department of Internal Medicine, Chang Gung Memorial Hospital, Linkou 333423, Taiwan; 7Circulating Tumor Cell Lab., Division of Hematology and Oncology, Department of Internal Medicine, Chang Gung Memorial Hospital, Linkou 333423, Taiwan

**Keywords:** non-small cell lung cancer, early-stage, circulating tumor cells, early detection, minimal residual disease’ mixed model repeated measures

## Abstract

Background: The role of circulating tumor cells (CTCs) for predicting the recurrence of cancer in lung cancer patients after surgery remains unclear. Methods: A negatively selected protocol of CTC identification was applied. For all the enrolled patients, CTC testing was performed before and after surgery on the operation day (day 0), postoperative day 1, and day 3. The daily decline and trend of CTCs were analyzed to correlate with cancer relapse. The mixed model repeated measures (MMRM) adjusted by cancer characteristics was applied for statistical significance. Results: Fifty patients with lung mass undergoing surgery were enrolled. Among 41 primary lung cancers, 26 (63.4%) were pathological stage Tis and I. A total of 200 CTC tests were performed. MMRM analysis indicated that surgery could contribute to a CTC decline after surgery in all patients with statistical significance (*p* = 0.0005). The daily decrease of CTCs was statistically different between patients with and without recurrence (*p* = 0.0068). An early rebound of CTC counts on postoperative days 1 and 3 was associated with recurrence months later. Conclusion: CTC testing can potentially serve as a tool for minimal residual disease detection in early-staged lung cancer after curative surgery.

## 1. Introduction

Lung cancer is one of the leading causes of cancer-related deaths worldwide [1]. According to the National Comprehensive Cancer Network guidelines, the standard management for resectable non-small cell lung cancer is anatomic resection with mediastinal lymph node dissection, followed by adjuvant therapy if needed [2]. For patients with lung cancer who present with resectable disease, the major factor that affects patients’ survival is recurrence. A literature review revealed that many relapse risk factors have been identified from clinicopathologic characteristics, and prediction models have been proposed [3,4,5,6]. These findings have been used to identify high-risk patients and for the planning of personalized surveillance programs. However, the detection of disease recurrence still relies on conventional imaging studies, including positron emission tomography (PET) and computed tomography (CT) [2]. PET can measure glucose metabolic activity [7,8] and reveal potential hot spots with increased glucose consumption [9]. CT can show the actual anatomic shape and size of the lesion(s), but comparisons among longitudinal imaging studies are needed for interpretation. Although the sensitivity of imaging technology has improved much in the past decades, several limitations to imaging studies still exist.

One of the limitations of the PET scan is the difficulty in differentiating lesions that are less than 1 cm from a lesion with inflammatory etiology [10]. Regarding CT scan, the baseline imaging settings, including slice thickness and radiation setting, could affect the presentation of the lesion(s) with suspicion of malignancy [11,12]. Although the RECIST criteria [13] have been used as standards for response evaluation of tumors, individual variations exist among different radiologists. The most crucial disadvantage of these imaging modalities is that they only detect the space-occupying lesions but not the minimal residual disease, i.e., disease status without image-visible lesions. In addition, many undetermined lung lesions are being identified by lung cancer screening, and tissue proof is crucial for lesions with a suspicion of malignancy. The timing of tissue proof for a lesion with suspicion of malignancy has been discussed extensively [2,14], but the treatment plan still depends on imaging tools. For surveillance in patients with lung cancer, close imaging studies with the risk of radiation over-exposure or repeated biopsy with possible procedure-related complications are not the best policy. We urgently need a less invasive but more sensitive tool to help us precisely detect minimal residual disease (MRD) in order to improve patients’ survival.

Successful tumor relapse requires many pathophysiologic cascades, including loss of cellular adhesion, increase in cancer cell motility and invasiveness, entry into and survival in the circulation, extravasation into the surrounding tissue, arrest in a new organ, initiation of growth, and vascularization of the metastatic tumor(s) [15,16]. In these essential processes of cancer expansion, growth, and metastasis, circulating tumor cells (CTCs) might represent an important phase [17]. CTCs are rare but have the capacity to metastasize [17,18]. Many studies have been conducted to clarify the role of CTCs. From the literature review, CTCs could be identified in all patients with lung cancer [19,20]. For early-stage lung cancer, thyroid transcription factor-1 (TTF-1) positive CTCs could be detected from peripheral blood and correlated with poor prognosis and a shorter progression period [21,22]. For patients with proven lung cancer with distant metastases, a higher number of CTCs is related to poor prognosis [23]. Hence, CTC testing could be utilized as a monitoring tool in advanced lung cancer patients to identify the changes in tumor dynamics during treatment [24]. The change in CTC numbers could be used as a tool for treatment response evaluation in advanced lung cancer patients [25]. However, no study has been conducted to perform CTC tests longitudinally after curative surgery and compare CTC counts with their baseline values. The possible role of CTC testing in predicting recurrence in patients with pulmonary malignancy remains debatable. In this prospective observational study, we routinely performed CTC testing before and after curative surgery. We hypothesized that CTCs could serve as one tool of MRD and might predict the recurrence several months after curative surgery.

## 2. Materials and Methods

### 2.1. Patients and Criteria of Enrollment

This study was conducted in a single medical center and approved by the Institutional Review Board (IRB) of Chang Gung Memorial Hospital at Linkou, Taiwan, with approval IDs of 201508753B0 and 201701892B0. Patients with resectable lung lesion(s) with suspicion of malignancy were enrolled. We explained the benefits and risks of the trial before enrollment and obtained informed consent from each patient. Eligible patients were those who presented with resectable pulmonary nodules with suspected malignancy. Other inclusion criteria included (1) age above 20 years; (2) capacity to fully understand and grant informed consent; (3) completion of all images related to resectability prior to surgery; and (4) agreement to the blood drawing schedule. The study protocol (Figure 1) has been registered at www.ClinicalTrials.gov with the protocol number of NCT03724500. Patients with a recent cancer history within 3 years were excluded at enrollment.

### 2.2. Pre-Operative Evaluation

A complete imaging survey was performed for patients who presented with resectable pulmonary nodules with suspicion of malignancy prior to operation. A chest CT scan was performed to identify tumor location and mediastinal lymph node invasion status. PET-CT scan was performed to rule out extrapulmonary metastatic lesions. Both exams were conducted for resectability evaluation. A pulmonary function test was performed for patients’ pulmonary reserve evaluation. Cardiac echo was performed for older patients and those who presented with multiple comorbidities, such as hypertension, diabetes, and chronic pulmonary obstructive disease, to identify the cardiac function. These two exams were conducted for cardiopulmonary reserve evaluation. Only patients presenting with the resectable disease were included.

### 2.3. Operational Methods and Post-Operative Treatment

Tissue proving was completed first before tumor resection. For peripheral lesion, wedge resection was used to obtain tumor tissue for diagnosis confirmation. If a benign lesion was confirmed, no further mediastinal lymph node dissection was performed. If the diagnosis of malignancy was established, the following operational procedures depended on the patient’s cardiopulmonary reserve. For patients with a normal cardiopulmonary reserve, complete lobectomy or segmentectomy with mediastinal lymph node dissection was performed as surgery with curative intent. In patients with compromised cardiopulmonary reserve, sub-lobar resection, i.e., wedge resection or segmentectomy with mediastinal lymph node dissection was performed.

For central lesions, lobectomy or segmentectomy was used to obtain tumor tissue for diagnosis confirmation. If a benign lesion was confirmed, no further mediastinal lymph node dissection would be performed. If malignancy was confirmed, the following operational procedures depended on the patient’s cardiopulmonary reserve. For patients with healthy cardiopulmonary reserve function, lobectomy or segmentectomy with mediastinal lymph node dissection was performed as surgery with curative intent. For patients with compromised cardiopulmonary reserve, segmentectomy with mediastinal lymph node dissection was performed. The decision algorithm for operations is shown in Appendix A.

No further treatments were administered to patients with confirmed benign lesions. For those who presented with infectious disease, anti-infection treatment was given according to culture and pathology results. For patients who presented with primary pulmonary malignancy, postoperative adjuvant therapy was given according to the TNM staging system. Further palliative treatment was administered to patients who presented with extrapulmonary malignancy with lung metastases.

### 2.4. Surveillance and Management of Suspicious Relapse

All patients were scheduled for regular quarterly follow-ups in the out-patient department. Chest CT was used as a surveillance imaging tool during the follow-up visits. The CTCs were re-checked when CT was performed. The response evaluation of tumor status by CT was based on RECIST criteria version 1.1 [13] with a cutoff value of CTCs at 3.0 cells/mL, as used in previous literature [26,27]. If CT revealed suspected relapsed lesions, repeated PET-CT and brain MRI studies were performed for differential diagnosis from some inflammatory conditions. For those likely to be cancer relapses or presenting with multiple lesions, surgical or image-guided biopsy for tissue proof was used to decide the treatment plan. For those with resectable and solitary lesions with a suspicion of malignancy, the tumor was resected. Otherwise, image-guided biopsy for tissue proof was performed for those with unresectable lesions, even if they were solitary. For patients with no visible lesions after imaging but with greater than 3 cells/mL CTC counts, CTC level was first rechecked for confirmation. The management algorithms are shown in Appendix A.

### 2.5. Circulating Tumor Cell Sampling, Identification, and Quality Controls

For each patient, 4 mL peripheral blood samples were drawn before and right after surgery, on postoperative day 1 (POD1) and day 3 (POD3), as illustrated in Figure 1. In each sampling, the first 2 mL of blood was discarded in order to avoid skin epithelial cell contamination. Then 4 mL of peripheral blood was collected for CTC enumeration and identification using a validated and published protocol, which combined negative and positive detection strategies [28]. The negative selection included depletion of erythrocytes and leukocytes, in order to enrich CTCs. Erythrocytes were lysed within 24 h after sampling, followed by CD45 depletion, which was achieved by adding 25 μL/mL EasySep CD45 Depletion Cocktail (STEMCELL Technologies Inc., Vancouver, BC, Canada) and 50 μL/mL EasySep Magnetic Nanoparticles (STEMCELL Technologies Inc., Vancouver, BC, Canada). Epithelial cell adhesion molecule (EpCAM) immunostaining (1:300, STEMCELL Technologies Inc., Vancouver, BC, Canada) was used for positive selection and Hoechst 33342 (1:500 in washing solution; Thermo Scientific, Waltham, MA, USA) for nuclear staining. CTCs were defined as cells that were negative for CD45 and positive for both EpCAM and Hoechst. We set the cutoff at 3.0 cells/mL as used in the literature to avoid false positives [29,30].

Flow cytometry using the CytoFLEX Flow cytometer (Beckman Coulter, San Diego, CA, USA) quantitatively identified and counted CTCs. An EpCAM isotope was used for primary controls. In addition, 4 mL peripheral blood drawn from healthy individuals were spiked with and without 1000 A549 cells (lung cancer cell line obtained from the Food Industry Research and Development Institute, Hsinchu, Taiwan) for the secondary control. Two kinds of controls were used routinely to avoid pseudo-positive and pseudo-negative CTC identification as standard quality controls of CTC testing. The performance of recovery, which was defined as the number of A549 cells detected by flow cytometry divided by the number of spiked A549 cells, and the coefficient of variation (CV) value were calculated and reported as stable in a previous report [31]. Briefly, the platform can achieve a recovery rate of 44.6% ± 9.1% and a percentage coefficient of variation (CV) of 20.4% [28].

### 2.6. Statistical Analysis

Descriptive statistics for continuous variables are expressed as mean ± SD and categorical variables as numbers (percentages). Differences in CTC numbers at baseline among subgroups were tested with the One-way Analysis of Variance (One-way ANOVA) test for continuous variables and the chi-square test for categorical variables. To assess the longitudinal trend of CTC numbers from pre-operative to postoperative day 3, we estimated the CTC daily decline rate by mixed model repeated measures (MMRM). Each CTC number was set as the dependent variable, whereas time by day was set as the independent variable. CTC daily decline rate was defined as the β coefficient of time by day in MMRM. Interactions among CTC daily decline, gender, benign or malignant status, pathologic stage, and clinical scenarios were tested and adjusted by cancer characteristics, including cancer staging, age, and gender. The crude and adjusted CTC daily decline rates by MMRM models were reported separately. All reported p values were two-sided and considered significant when *p* < 0.05. In addition, 95% confidence intervals are reported. All statistical analyses were performed using SPSS version 18.0. 

## 3. Results

### 3.1. Patient Enrollment

From January 2017 to December 2018, 51 consecutive patients were enrolled in this study. One patient who could not tolerate tumor resection due to unstable intraoperative saturation was excluded. The study design included a 3-year follow-up by routine CTC testing (please refer to the design of the NCT03724500 trial at www.ClinicalTrials.gov). Herein, we reported the CTC trends within the first 3 days after surgery. The details of the CTC testing within 3 days of surgery are illustrated in Figure 1. The mean age of the participants was 63.0 years (ranging from 31.0 to 91.0 years). Table 1 demonstrates the basic characteristics of enrolled patients according to the intent-to-treat principle (N = 50). Most patients had good clinical pretreatment performance statuses (96.1% with Eastern Cooperative Oncology Group (ECOG) performance status 0). Most patients (92.0%, 46/50) received a standard anatomic resection with mediastinal lymph node dissection. The remaining 8.0% underwent wedge resection because of the poor pulmonary reserve. The most common surgery methods included lobectomy with mediastinal lymph node dissection (58%) and segmentectomy with mediastinal lymph node dissection (34.0%). Almost all patients had normal serum carcinoembryonic antigen (CEA) or squamous cell carcinoma (SCC) levels before surgery.

### 3.2. Circulating Tumor Cell Counts

After a median follow-up time of 12.1 months (range, 1.5–23.6 months), a total of 200 CTC tests were performed as trial scheduled before December 2018 (data cutoff date). The mean and median CTC numbers for the whole cohort were both 7.0 cells/mL (range, 0.0–74.7 cells/mL; standard deviation (SD), 2.4 cells/mL) (Table 2). For patients with benign etiology, the CTC number was 1.0 cell/mL (range 0.0–6.0 cells/ mL). Patients who presented as lung metastases from extrapulmonary malignancy had a mean CTC number of 7.5 cells/mL (range 3.0–74.5 cells/mL). The mean CTC number for whole lung cancer patients and those conformed with relapse was 8 cells/mL (range 0.0–74.7 cells/mL) and 6.5 cells/mL (range 0.5–42.5 cells/mL), respectively. There was no significant difference among these groups (*p* = 0.49).

### 3.3. Subgroups Based on Pathologic Findings and Cancer Outcome (Cancer Relapse)

In the cohort, although primary pulmonary malignancy was suspected after standard examinations with PET/CT and chest CT scans, three different diagnoses were confirmed after surgery: (i) benign (n = 4), including two cryptococcal infections, one pneumonia, and one necrotizing granuloma; (ii) primary lung malignancy (n = 41); and (iii) oligo-metastasis, by special pathology immunostaining markers and past cancer history post-curative treatment (n = 5). These five patients included two breast cancers, two colon cancers, and one cervical cancer. They were not initially excluded at the enrollment because their cancer had been curatively treated by surgery and had no sign of relapse within 3 years. At the data cut-off date in December 2018, 11 patients had experienced a lung cancer recurrence.

We subsequently correlated and compared the pathologic features, and the median CTC counts in each subgroup (Table 3). In this study, most patients (82.0%, 41/50) were pathologically diagnosed with lung cancer. Five patients presented with extrapulmonary malignancy with solitary lung metastasis, whereas four patients presented with benign disease. Given the cancer recurrence status confirmed at analysis, four different clinical scenarios were identified, including primary lung cancer with (n = 11) and without relapse (n = 30), benign group (n = 4), and extrapulmonary malignancy with lung metastasis (n = 5). Pathologic classification of lung cancer showed stage 0 (4.0%), I (48.0%), II (12.0%), III (16.0%), and stage IV (1.0%).

### 3.4. The Positivity of Circulating Tumor Cell Detection in Different Clinical Conditions

Figure 2 demonstrates the positivity rate of CTC detection within 3 days after surgery. The median CTC counts before curative surgery for primary lung cancer, the benign group, extrapulmonary malignancy with lung metastasis, and lung cancer with relapse were 41.0, 5.0, 6.0, and 11.0, respectively. Among patients who presented with extrapulmonary malignancy with lung metastasis, 80.0% were identified with CTC after surgery, whereas persistent CTCs were noted in 40.0% of patients while that in the benign group was 0.0%. For lung cancer without relapse until the data cut-off date (n = 30), CTCs could be identified in 74.2% of patients. The mean baseline (pre-operative day 0) CTC counts did not have a significant difference among the four entities (Figure 2). Interestingly, the mean CTC counts all dropped to the lowest level on postoperative day 1 among the four groups in this cohort. The mean CTC counts rebounded in the relapse group (n = 11) and oligometastasis group (n = 5) (Figure 3 and Figure 4). The rebound of circulating tumor cells might predict recurrence. In the standard surveillance period, 11 cases were pathologically proven to have recurrence from lung cancer. Two representative cases with continuous longitudinal monitoring after surgery, along with disease changes, were displayed in Appendix A.

### 3.5. Mixed Model Repeated Measures Analysis for Dynamic Changes of Circulating Tumor Cell Counts

Table 4 demonstrates the mixed model repeated measures (MMRM) analysis for CTC changes in each patient. First of all, CTC counts decreased with statistical significance in all patients. No differences were observed between the sexes. The decreased CTC count in the cancer group for the first 3 days was 2.3 cells/mL/day, compared to the benign group, which was statistically significant (*p* = 0.0009). The daily decline of CTC counts in lung cancer without recurrence (n = 30) was 2.5 cells/mL/day within the first 3 days of surgery and 1.4 cells/mL/day in lung cancer with recurrence (n = 11). For lung cancer without relapse, the daily decline of CTC counts was statistically significant after adjustment by age, sex, and staging (*p* = 0.0068). The CTC decline in stage I and Tis group (n = 26) was 2.9 cells/mL/day and remained statistically significant after adjustment (*p* = 0.0079). From the view of CTC variation in different clinical scenarios, the variation trends continued to decline without an upward rebound in the “benign” group and “lung cancer without relapse” group. In patients with extrapulmonary malignancy with lung metastasis (”oligometastasis” group) and “lung cancer with relapse” group, the CTCs declined to the lowest level on postoperative day 1 and increased again after postoperative day 3 (Figure 3). The *p*-value for interaction showed no statistical significance, which means that intergroup differences cannot be identified (Table 4). The findings may indicate a disseminated cancer condition and require adequate systemic therapy after the operation.

## 4. Discussion

With lung cancer screening becoming more widespread, much more undetermined lung lesions are being detected. In current practice, lesions presenting with size month intervals than 1 cm or increased consolidation tumor ratio (C/T ratio) are a target for tissue proof considering a high possibility of primary lung cancer [32]. However, variations in radiation dosage and slice thickness of imaging studies may result in changes in the presentation of tumor image and lead to difficulty and bias during the serial comparison [11,12]. Theoretically, three clinical scenarios could be encountered, including benign lesions, primary lung cancer, and extrapulmonary malignancy with lung metastasis or second primary lung cancer. In benign lesions, inflammation and fibrosis may lead to DNA damage and cause carcinogenesis [33,34]. In primary lung malignancy, the most common cell type is adenocarcinoma [2]. In patients with extrapulmonary malignancy, pulmonary ground-glass opacity lesion carries a higher risk of secondary pulmonary malignancy [35]. Among these clinical scenarios, primary pulmonary malignancy is a major concern, and tissue proof or resection is recommended if images change in the serial follow-up. In patients who underwent tumor resection and were confirmed with primary pulmonary malignancy, the risk of disease relapse was the major concern whereby the reported early relapse in patients with resectable non-small cell lung cancer is 10–19% [5,29]. Many prognostic factors have been identified but can only be used for surveillance planning [3,4,5], instead of relapse prediction [30].

CTCs have been widely investigated, and their correlation to survival has been shown [18,19,20,21,22,23,24,25]. They have been identified in the pulmonary vein during manipulation but also found in peripheral venous blood [31,36]. Crosbie et al. found that CTCs obtained from the pulmonary vein prior to manipulation and from peripheral venous blood before surgery were all correlated to relapse [37]. However, while the relapse prediction capacity of CTCs remains unknown, we have identified that CTCs decline in all patients suggests that surgery can efficiently decrease the tumor burden even in metastatic settings. In patients with lung cancer, we have identified that CTCs can be detected in a very early-stage lung cancer population, similar to that of Reddy et al.’s findings [38]. In our study, 26 patients (26/41, 63.4%) had early-stage lung cancer, including adenocarcinoma in situ and stage I disease. For patients with lung cancer with relapse, the lower declining slope and daily decline of CTC counts suggest cancer recurrence compared with higher CTC decline slope in patients without recurrence. For extrapulmonary malignancy with lung metastases, the CTCs decline after tumor removal and rebound to a higher level after postoperative day 3 even though on imaging, the visible lesion is completely resected. The finding suggests that occult metastases may exist in these scenarios and systemic therapy is warranted for better disease control. Our study shows that CTC not only could be identified in the early stage of the disease but also indicates their predictive capacity for disease relapse. To the best of our knowledge, this is the first study to suggest a serologic cancer remission in some cases, although this is not one of the pre-designed endpoints in the original research. Further prospective larger-scale studies are warranted.

From the view of CTC detection, high detection rates were identified in all malignant clinical scenarios. In this study, all CTCs were defined and selected by CD45^neg^, Hoechst^pos^, and Epi-CAM^pos^ using flow cytometry after negative selection processes. In cases with extrapulmonary malignancy with solitary lung metastasis, CTCs could be identified in all patients (100.0%) in our cohort. However, false alerts, i.e., elevated CTC counts greater than 3.0 cells/mL without evidence of malignancy in conventional imaging studies, were also noted during surveillance. In this cohort, we found a false alert rate of approximately 5.9% in this study. Baseline false-positive results may be related to epithelial contamination during peripheral blood sampling because these cells share the same surface antigen characteristics [39]. A possible reason for false-negative results may be related to occult cancer activity, in which cells lose common epithelial differentiation, i.e., epithelial–mesenchymal transition (EMT) [40] and the nature of dynamic equilibrium between developing cancer cells and the immune system [41]. Further investigation is warranted to minimize false-negative detection. 

Circulating tumor DNA (ctDNA) is an alternative biomarker for detecting MRD [42]. It originates as a breakdown product of dead CTCs or by active secretion of tumor cells [43]. However, the ctDNA amount is extremely low and it is renewed quickly, leading to difficulty in differentiating it from the background [42,44]. CTCs, by contrast, are tumor cells within the circulation that not only exhibit morphological characteristics but also allow immunohistochemical staining. In addition, their biological features, including EMT [40] or cancer stemness [45] may be correlated to tumorigenic or metastatic phenotype and resistance to therapies. Therefore, CTCs can provide more biological information than ctDNA. In a previous study on ctDNA by Chaudhuri et al., ctDNA was used to identify MRD after treatment in patients with locally advanced lung cancer and to clarify disease relapse 5.2 months earlier than regular image surveillance [46]. The applicability of this finding was limited by the study population and timing of blood sampling [47], and the relapse prediction power before treatment remains unclear. From the literature review, CTCs could be identified in all patients with lung cancer [19,20]. The correlation to prognosis [21,22,23,24,37] and the application of treatment response was also identified [20]. However, no longitudinal surveillance data has been gathered to clarify the relapse prediction power and test the possibility of clinical application. In this study, we did show that the daily decline of CTC counts in the first 3 days after surgery was correlated to disease relapse. Our findings could be used not only for MRD detection after the operation but also to detect microscopic relapse in regular surveillance.

Even though we clarified the role of CTCs in relapse prediction and detection, limitations remain. First, the number of participants was small. However, the majority of patients in this prospective study were patients with early-stage pulmonary adenocarcinoma, which was very different from previous studies [37,46]. Second, the post-operative surveillance period was short and the long-term result was still under investigation. However, our results indicated that daily decline of CTC counts could predict disease relapse, and may be integrated into post-operative treatment and surveillance planning. Third, the low false alert rate (5.9%) cannot be ignored, limiting CTC counts in clinical practice. Further investigation is warranted. Concurrently, high detection rates were recorded in all scenarios, especially for relapse detection. This finding suggests that CTCs could serve as a non-invasive diagnostic tool that could be applied for relapse detection. Further patient recruitment and longer longitudinal surveillance are needed to clarify the clinical significance of CTCs for patients with suspected malignancy.

## 5. Conclusions

CTCs showed a high detection rate in all clinical scenarios that may be encountered in resectable undetermined lung lesions. CTC counts drop to the lowest point on postoperative day 1 in all situations and rebound in postoperative day 3 for those patients with confirmed relapse. Greater CTC count decline in the first three days was identified in patients who presented with benign lesions and non-small cell lung cancer without relapse. In conclusion, we found that CTC testing can potentially serve as a tool for minimal residual disease detection in early-staged lung cancer after curative surgery.

## Figures and Tables

**Figure 1 diagnostics-10-00144-f001:**
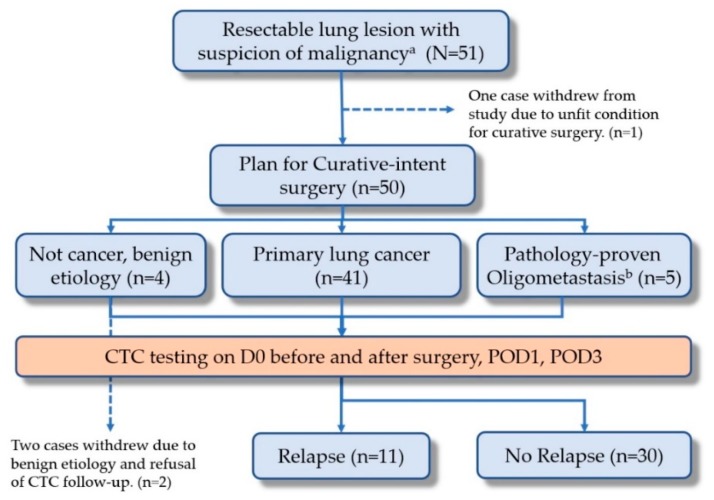
The study flow of the prospective design. (CTC, circulating tumor cell; POD1, postoperative day 1; POD3, postoperative day 3).

**Figure 2 diagnostics-10-00144-f002:**
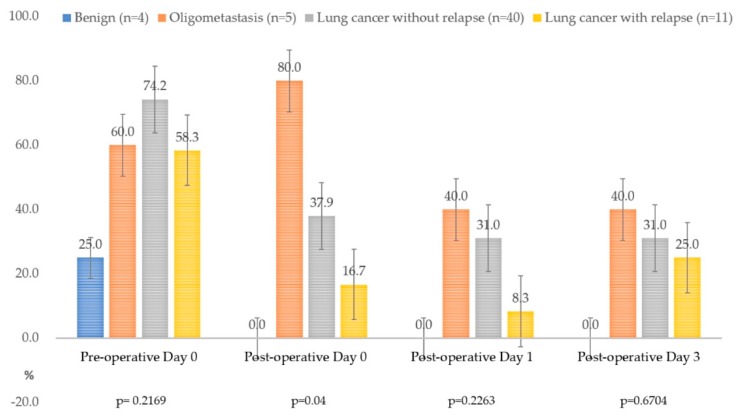
The positivity of circulating tumor cells (CTCs) in different clinical scenarios. The CTC positivity is not different among four subgroups before curative surgery and is statistically different immediately after the surgery (postoperative day 0, *p* = 0.04). A possible false positivity of 25.0% can be observed in the benign group (n = 4), but it goes down to zero after surgery. On the other hand, the CTC positivity of the oligometastasis group (n = 6) remains 40.0–80.0% though the operation has removed even the visible tumor(s).

**Figure 3 diagnostics-10-00144-f003:**
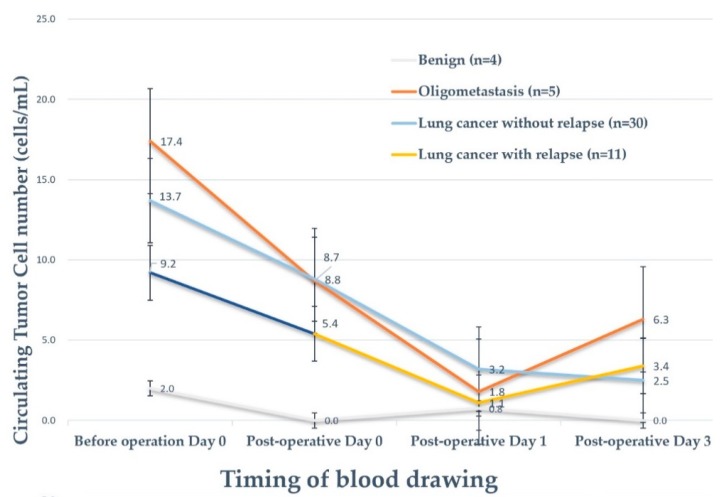
The rebound of circulating tumor cell counts correlates to minimal residual disease. In this figure, a rebound of mean CTC number was observed on day three after surgery in both oligometastasis group (red) and primary lung cancer with recurrence group (green), indicating possible minimal residual tumor signals in blood after surgery.

**Figure 4 diagnostics-10-00144-f004:**
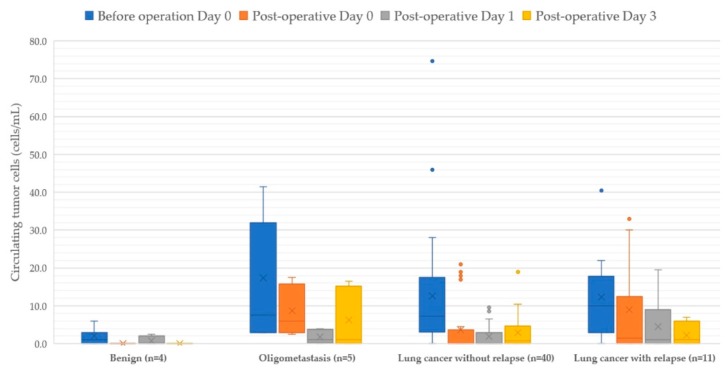
Boxplots of circulating tumor cells within 3 days of surgery.

**Table 1 diagnostics-10-00144-t001:** Basic characteristics of enrolled patients. ECOG, Eastern Cooperative Oncology Group; CEA, Carcinoembryonic antigen; SCC. squamous cell carcinoma; CTC: Circulating tumor cells.

Clinical Characteristics	N	Range/%
Age, median (years)	63.0	31.0–91.0
Sex, male/female	26/24	52.0%/48.0%
ECOG performance status		
0	49	98.0%
1	1	2.0%
Smoking history		
Never	34	68.0%
Former or Current	16	32.0%
Operational method		
Lobectomy + mediastinal lymph node dissection	29	58.0%
Segmentectomy + mediastinal lymph node dissection	17	34.0%
Wedge resection + mediastinal lymph node sampling	4	8.0%
Pre-operative lab data	Value	Reference values
CEA (ng/mL), reference range	2.8 ± 3.2	< 5.0
SCC (ng/mL), reference range	1.0 ± 0.6	< 2.5
Albumin (g/dL)	4.3 ± 0.3	3.5–4.5
Hemoglobin (gm/dL)	13.4 ± 1.6	M: 13.5–17.5; F: 12–16
Hematocrit (%)	40.2 ± 4.51	M: 41–53; F: 36–46
WBC (cells/μL)	8922.0 ± 16,418.0	M: 3900–10,600
Segment (%)	60.3 ± 10.9	42.0–74.0
Monocyte (%)	6.1 ± 2.2	0.0–12.0
Lymphocyte (%)	29.5 ± 10.9	20.0–56.0
Duration of Follow-up (months), Median (range)	12.1	(1.5–23.6)

**Table 2 diagnostics-10-00144-t002:** Circulating tumor cell counts in all patients and different groups. CTC, circulating tumor cells.

Baseline CTC Counts	N	Median (Range)	*p* Value
CTC counts (cells/mL), All patients	200	5.6 (0.0–74.7)	
CTC counts, subgroups			
Primary lung cancer group	41	8.0 (0.0–74.7)	0.49
Benign group	4	1.0 (0.0–6.0)	
Oligometastasis from other cancer	5	7.5 (3.0–74.5)	
Recurrence group from lung cancer	11	6.5 (0.5–42.5)	

**Table 3 diagnostics-10-00144-t003:** Pathology characteristics of the study cohort (N = 50). AJCC, American joint committee on cancer; NA, not available.

Diagnosis/Etiology of Lung Nodule	N	(%)
Primary lung cancer	41	(82.0%)
Not primary lung cancer	9	
Metastatic group ^a^	5	(10.0%)
Benign group ^b^	4	(8.0%)
Lung cancer staging (AJCC 8th edition)	41	
Carcinoma in situ	2	(4.0%)
Ia1/Ia2/Ia3	6/8/6	(12.0/16.0/12.0%)
Ib	4	(8.0%)
IIa	1	(2.0%)
IIb	5	(10.0%)
IIIa	8	(16.0%)
IV ^c^	1	(2.0%)
Pathologic characteristics (lung cancer), n = 41		
Tumor size, (cm)	2.24 ± 1.28	
G1/G2/G3/NA	17/13/7/4	(41.5%/25.5%/13.7%/7.8%)
Pathology (lung cancer), n = 41		
Adenocarcinoma	33	(80.5%)
Invasive mucinous adenocarcinoma	3	(7.3%)
Squamous cell carcinoma	3	(7.3%)
Inflammatory myofibroblastic tumor	1	(2.4%)
Lymphoepithelioma-like carcinoma	1	(2.4%)

^a^ This entity includes two breast cancers, two colon cancers, and one cervical cancer. ^b^ Benign etiology consists of two cryptococcal infections, one pneumonia, and one necrotizing granuloma. ^c^ Patients with Stage IV include 1 stage IVa with only pleural metastasis from lung adenocarcinoma. Oligometastases were excluded here.

**Table 4 diagnostics-10-00144-t004:** Crude and adjusted daily decline among different subgroups.

Group	N	Crude Daily Change of CTC in 3 Days	95%CI		*p*-Value	Adjusted Daily Change of CTCs in 3 Days	95%CI		*p*-Value	*p*-Value for Interaction
Total	50	−2.1129	−3.2998	−0.9259	0.0008	−2.1262	−3.2806	−0.9718	0.0005	
Sex										
Male	28	−1.9206	−3.0995	−0.7416	0.0025	−1.9397	−3.0756	−0.8038	0.0017	0.7046
Female	22	−2.3692	−4.6379	−0.1006	0.0415	−2.3692	−4.6228	−0.1156	0.0403	
Malignancy vs. Benign										
Benign	4	−0.3786	−1.1618	0.4047	0.2216	−0.3786	−1.1566	0.3995	0.2193	0.3665
Malignant	46	−2.2637	−3.5444	−0.9831	0.0009	−2.2803	−3.5361	−1.0245	0.0007	
Metastasis vs. Benign										
Benign	4	−0.3786	−1.1618	0.4047	0.2216	−0.3786	−1.1566	0.3995	0.2193	0.8660
Metastatic	5	−2.5543	−6.8846	1.7761	0.1768	−2.5543	−6.2984	1.1899	0.1311	
Lung cancer without recurrence	30	−2.5359	−4.3652	−0.7065	0.0083	−2.5702	−4.3728	−0.7676	0.0068	
Lung cancer with recurrence	11	−1.4000	−3.1405	0.3405	0.1033	−1.4000	−2.9854	0.1854	0.0775	
Stages										
Benign entity	4	−0.3786	−1.1618	0.4047	0.2216	−0.3786	−1.1566	0.3995	0.2193	0.8242
Stage I and Tis	26	−2.8526	−4.8985	−0.8068	0.0082	−2.8526	−4.8892	−0.8161	0.0079	
Stage II	6	−1.5586	−4.2542	1.1371	0.1974	−1.5586	−4.1012	0.9840	0.1759	
Stage III	8	−0.07411	−1.9415	1.7932	0.9258	−0.08582	−1.3883	1.2167	0.8772	
Stage IV	6 ^a^	−3.1952	−6.7311	0.3406	0.0678	−3.1952	−6.6038	0.2133	0.0609	

^a^ Stage IV group included 5 metastatic cancers and one stage IVc lung cancer. CTC, circulating tumor cells; CI, confidence interval.

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
