# Peer review of "Circulating Tumor Cells as a Tool of Minimal Residual Disease Can Predict Lung Cancer Recurrence: A longitudinal, Prospective Trial"

_diagnostics, 2020, doi:10.3390/diagnostics10030144_

Round 1

Reviewer 1 Report

see attached file

Reviewer 2 Report

The manuscript demonstrates that following surgery, CTC's increase in lung cancer patients who experience relapse compared those who do not experience relapse/recurrence, comparing days one to three post-resection.  The statistical analyses and methodology are valid. Although the sample number is a little low, the study is valid and gives new information concerning CTC's and lung cancer recurrence following surgical resection.

I would like to see one aspect of the study addressed. CTC''s were detected in patients without lung cancer. The manuscript states that these cells might be contaminants secondary to the biopsy procedure. It would help if this issue was addressed by looking for CTC's in patients without cancer and thereby "pin down" the phenomenon of CTC's in patients without cancer. Establishing a clearly defined false positive rate from such patients would improve an otherwise excellent study. This should be addressed. 

Round 2

Reviewer 1 Report

I feel the authors have improved the manuscript and answered almost all my concerns.

Reviewer 2 Report

The paper has addressed small flaw in the study and further explained it an made appropriate corrections. The "circulating tumors cells" identified in the blood of individuals without cancer was explained. It's important that these two individuals had no circulating tumor cells after the first day of blood draws and the number of "rumor cells" in these individuals was low compared to those who had cancer. The small flaw in the first manuscript was dealt with.